# Effects of Nutritional Support with a Leucine-Enriched Essential Amino Acid Supplement on Body Composition, Muscle Strength, and Physical Function in Stroke Patients Undergoing Rehabilitation

**DOI:** 10.3390/nu16244264

**Published:** 2024-12-11

**Authors:** Naoki Nakagawa, Satoshi Koyama, Keisuke Maruyama, Jun-Ichi Maruyama, Naoyuki Hasebe

**Affiliations:** 1Division of Cardiology and Nephrology, Department of Internal Medicine, Asahikawa Medical University, 2-1-1-1 Midorigaoka-Higashi, Asahikawa 078-8510, Hokkaido, Japan; 2Department of Internal Medicine, Asahikawa Rehabilitation Hospital, Asahikawa 078-8510, Hokkaido, Japan

**Keywords:** branched-chain amino acid, rehabilitation, renal function, stroke, sarcopenia, vitamin D

## Abstract

Background/Objectives: Dietary protein intake can potentially influence renal function. This study aimed to elucidate the association between dietary protein supplementation and a decrease in the estimated glomerular filtration rate (eGFR) in Japanese stroke patients undergoing rehabilitation. Methods: From July 2017 to June 2021, 60 patients undergoing post-stroke rehabilitation were randomly assigned to a rehabilitation alone or rehabilitation nutrition group, which received 120 g Reha-Time Jelly^®^ after each session. Both groups were followed up for 3 months. Serum nutritional markers (prealbumin and retinol-binding protein), muscle strength, body composition, renal function markers (eGFR based on creatinine [eGFR-Cr] and cystatin C [eGFR-Cys]), urinary protein-to-creatinine ratio (UPCR), and motor function (walking speed, 2-min walk distance, and chair stand test) were assessed at baseline and post-intervention. Results: Of the 60 participants (mean age: 70.2 ± 10.0 years), 39 were men (65.0%) and 19 (31.7%) had chronic kidney disease. Initial eGFR-Cr and eGFR-Cys values were 70.5 ± 17.2 and 66.6 ± 14.8 mL/min/1.73 m^2^, respectively. After the intervention, the rehabilitation nutrition group demonstrated a significantly greater increase in body mass index (BMI) and a smaller decrease in bone mineral content than the rehabilitation alone group. However, no significant between-group differences were noted in serum marker levels or motor function, including grip strength and knee extensor strength, on the paralyzed and non-paralyzed sides. The change in chair stand test performance indicated a trend toward improvement in the rehabilitation nutrition group. No significant differences were observed in the changes in renal function. Conclusions: A 3-month nutritional supplementation intervention may help increase BMI, preserve bone mineral content, and support physical activity levels in patients undergoing post-stroke rehabilitation without negatively affecting renal function.

## 1. Introduction

Stroke is the leading cause of mortality worldwide, and many survivors face prolonged challenges in performing activities of daily living (ADLs) as a direct consequence of this condition [1]. More than two-thirds of stroke survivors require rehabilitation after hospital discharge [1]. Approximately 8.2–49% of patients with stroke experience malnutrition [2,3]. Malnutrition in these patients exacerbates disease severity, increases mortality rates and infection risk, and often leads to swallowing difficulties, ultimately impeding post-stroke ADL performance [4]. Therefore, comprehensive nutritional support is critical for stroke rehabilitation [5,6]. However, unlike other areas of stroke treatment, such as the establishment of designated stroke centers, symptom recognition, and rehabilitation service delivery, nutritional support has received limited attention in post-stroke care. Increasing evidence suggests that combining nutritional support and rehabilitation is fundamental for effective stroke management [7,8].

Sarcopenia, characterized by age-related declines in muscle mass, strength, and function, is a major contributor to physical frailty and a critical area of study in stroke research [9,10]. Its development and progression in stroke patients are complex, encompassing shifts in muscle fiber type, asymmetries in physical function, dysphagia with subsequent malnutrition, disuse atrophy, and systemic metabolic alterations, along with various comorbid and pre-stroke risk factors, such as advanced age, physical inactivity, and suboptimal nutritional status [8]. Factors affecting nutritional status, both pre- and post-stroke, are important, as they represent potentially modifiable risk factors for stroke-related sarcopenia. Furthermore, older age and being thinner, but not worse, renal function were associated with a higher prevalence of sarcopenia in older adults with diabetes [11]. Current recommendations for sarcopenia prevention and treatment emphasize adequate protein and vitamin D intake [12,13]. Given the reduced sensitivity of aging muscle tissues to low amino acid doses, leucine-enriched amino acid supplementation is recommended to enhance muscle protein synthesis [5,6]. Recent systematic reviews and meta-analyses have further substantiated the efficacy of combining exercise with nutritional interventions for sarcopenia management, particularly advocating resistance training to optimize outcomes [14].

We hypothesized that resistance training combined with timely administration of leucine-enriched essential amino acid supplementation would improve muscle mass, strength, and physical function in patients hospitalized after a stroke who have sarcopenia and are at a heightened risk of disability. This randomized controlled trial aimed to evaluate the efficacy of this intervention on muscle mass (appendicular muscle mass measured by bioelectrical impedance analysis), muscle strength (handgrip strength), and functional performance in ADLs, as well as its safety regarding renal function.

## 2. Materials and Methods

### 2.1. Design and Setting

This single-center, randomized, parallel-group study was conducted at Asahikawa Rehabilitation Hospital, Asahikawa, Japan, from July 2017 to June 2021, using a physician-initiated central registration system. The study protocol was approved by the Institutional Review Board and registered in the UMIN Clinical Trials Registry (UMIN000027939). The study procedures were performed in accordance with the ethical recommendations of the Declaration of Helsinki. All participants were fully informed of the study and written consent was obtained from them.

### 2.2. Participants

The participants were males and females aged ≥40 to <85 years who regularly visited Asahikawa Rehabilitation Hospital as stroke outpatients. The diagnosis of stroke was based on standard clinical and imaging findings. All patients had motor paralysis due to a history of stroke but could walk independently. Patients were excluded if they met any of the following criteria:(1)Myocardial infarction, unstable angina, or peripheral artery disease in the past 6 months(2)Cardiac insufficiency in the acute phase(3)Under treatment for a malignant tumor(4)Undergone surgery in the past 6 months(5)Allergy to milk and/or soybean(6)Judged inappropriate as a subject of the study by the investigator.

### 2.3. Registration Procedure

The study secretariat confirmed that the candidates met all inclusion criteria and did not meet any exclusion criteria based on case registration sheets prepared by attending physicians. Subsequently, the candidates were officially registered and randomly allocated to either an exercise group or an exercise + nutritional supplementation group using dynamic adjustment allocation (minimization method), with age and sex as stratification factors. The allocation results were reported to the investigators.

### 2.4. Protocol Treatment

After informed consent was obtained from the eligible patients, blood sampling was performed during the observation period (within 4 weeks). Patients were asked to perform exercise or exercise combined with nutritional supplement intake for 12 weeks. After these 12 weeks, blood sampling, blood pressure measurement, and physical measurements, including body composition, were performed for all participants.

### 2.5. Rehabilitation Program

Both groups received guidance on resistance exercise training in the rehabilitation program from a physical therapist [15,16]. The program was tailored to individuals based on their functional capacity and disability levels. The basic exercises included: (1) standing up from a sitting position (sit-to-stand exercise), (2) raising the right and left legs alternately to an angle of 60°without bending the knees while lying on the back of the floor (single-leg raise exercise), and (3) slowly raising the hips so that the abdominal region and knees were aligned in a straight line while lying on the back of the floor, holding the position for a moment, and then lowering the hips slowly (hip lift exercise). For the sit-to-stand exercise, the participants were instructed to use a handrail as needed.

Participants in the exercise group were instructed to perform resistance exercise training at least three times a week. A physical therapist confirmed that the exercises were performed during monthly visits. Participants in the exercise + nutritional supplement group were instructed to exercise in the same manner and were provided Reha-Time Jelly^®^ to consume within 1 h of completing resistance exercise. The branched-chain amino acid (BCAA)-enriched nutritional supplement used in this study was developed to enhance the effects of rehabilitation aimed at increasing muscle strength [17]. It has the advantages of being a crushed jelly that is easily swallowed, even by patients with mild dysphagia, contains no fat, and has a fresh taste, thereby making it easy to consume after exercise. The oral nutritional supplement used comprised 20 μg vitamin D, 10.0 g protein (BCAA 2500 mg and leucine 1400 mg), 60% carbohydrates, 0% lipids, and 40% protein, providing 100 kcal energy with a muscat flavor.

### 2.6. Evaluation of the Results

At screening, information was collected regarding sex, birth date, height, weight, complications, medical history, present history (e.g., diagnosis and complications), daily life habits (e.g., alcohol intake and smoking), previous treatment, and present treatment (combination of drugs). Participants visited the hospital once a month for blood pressure and pulse rate measurements. These were measured twice, with a ≥5-min rest interval while sitting, and the mean value was recorded. The pulse rate was measured for 15 s, and the value was quadrupled for recording. The participants were asked to measure their blood pressure at home daily during the study period, within 1 h of waking and before bedtime, and to record all values in a blood pressure notebook.

The following examinations were performed for all participants during observation and at study completion.

(1)General pharmacological tests (red blood cell count, white blood cell count, hemoglobin, hematocrit, platelet count, aspartate aminotransferase, alanine transaminase, gamma-glutamyl transpeptidase, alkaline phosphatase, lactate dehydrogenase, serum creatinine, estimated glomerular filtration rate [eGFR], uric acid, albumin, and C-reactive protein(2)Nutritional markers: geriatric nutritional risk index (GNRI) [18], serum prealbumin and serum retinol-binding protein(3)Urine examination (spot urine): urinary microalbumin and urine creatinine(4)Other markers (cardiac marker: NT-pro BNP, to consider potential cardiovascular comorbidities affecting the overall outcomes)(5)Electrocardiogram (12 leads)(6)Body composition (muscle and adipose mass) using InBody S10 (InBody, Tokyo, Japan) is a validated bioelectrical impedance analysis (BIA) instrument [19](7)Motor function (unaffected grasping power, knee extension strength, 10-m walking steps and velocity, 2-min walking distance, standing up from a sitting position on a chair [30 s]).(8)Functional Independence Measure (FIM) [15](9)Sarcopenia was diagnosed based on the Asian Working Group for Sarcopenia (AWGS) 2019 criteria of low muscle mass, low muscle strength, or low physical function [20]. Sarcopenia was defined as having a low appendicular skeletal muscle mass (ASM), <7.0 kg/m^2^ in men and <5.7 kg/m^2^ in women using BIA, with either low muscle strength (the cut-off for low muscle strength on the non-paralyzed side was <28 kg for males and <18 kg for females) or low physical performance (the cut-offs for low physical performance was a usual gait speed <1.0 m/s). Severe sarcopenia was defined as low ASM with both low muscle strength and low physical performance.

Subjective symptoms and objective findings were confirmed through interviews and physical examinations during hospital visits [15]. All adverse events, including cerebral and cardiovascular events, were recorded regardless of their causal relationship with the study intervention and rehabilitation. A blinded rehabilitation therapist evaluated the results. A physical therapist blinded to the group allocation assessed the physical condition of the participants at baseline and upon completion of the intervention.

### 2.7. Statistical Analysis

Baseline characteristics were compared between the two groups using the Student’s *t*-test, Mann–Whitney U test, and chi-square (χ^2^) test. Post-intervention characteristics were assessed using analysis of covariance, with baseline characteristics included as covariates. *p*-values, means, and 95% confidence intervals (CIs) were calculated. Group differences in changes from baseline were analyzed using the Mann–Whitney U test. A two-sided *p*-value of <0.05 was considered statistically significant. All statistical analyses were performed using SPSS version 26 (IBM Corp., Armonk, NY, USA).

## 3. Results

### 3.1. Study Participants

We screened 65 post-stroke patients for eligibility, 60 of whom were randomly assigned to either the rehabilitation alone or the rehabilitation + nutrition groups. Five participants refused to continue the study and withdrew. The final rehabilitation alone and rehabilitation + nutrition groups consisted of 31 (median age, 72 years) and 29 patients (median age, 70 years), respectively.

### 3.2. Baseline Characteristics

The baseline characteristics of the patients are summarized in Table 1. None of the items examined differed significantly between the two groups at baseline. The median age of the entire study cohort was 71 years, and most patients were men (65.0%). The markers and motor function were not significantly different between the two groups. Intervention compliance was 100% from baseline to follow-up for all patients in both groups.

Approximately half of the enrolled patients with stroke had sarcopenia. The prevalence of sarcopenia and severe sarcopenia was 51.6% and 16.1%, and 58.6% and 20.7% in the rehabilitation alone and rehabilitation + nutrition groups, respectively (Table 1).

The nutritional marker, GNRI, was 103.2 ± 3.7 and 102.9 ± 4.7 in the rehabilitation alone and rehabilitation + nutrition groups, respectively (Table 1). Furthermore, the urine protein−creatinine ratio was very low in both groups, suggesting that the baseline nutritional status was relatively well maintained in both groups.

### 3.3. Primary Outcomes

After the intervention, markers and motor function were not significantly different between the two groups. Notably, regarding changes in body composition, the rehabilitation + nutrition group showed a significantly greater increase in body mass index (+0.24 ± 0.65 vs. −0.06 ± 0.48 kg/m^2^, *p* < 0.05) and a significantly smaller decrease in bone mineral content (−0.00 ± 0.01 vs. −0.06 ± 0.12 kg, *p* < 0.05) than the rehabilitation alone group (Figure 1).

Although grip strength and knee extensor strength on both the paralyzed and non-paralyzed sides did not differ significantly between the two groups, the degree of change in the chair stand test results showed a trend toward improvement in the rehabilitation + nutrition group (+1.3 ± 2.8 kg) compared with that in the rehabilitation alone group (+0.4 ± 2.1 kg) (Table 2).

### 3.4. Secondary Outcomes

No significant differences were observed in the changes in GNRI, eGFR-Cr, eGFR-Cys, or UPCR between the two groups (Table 2). Additionally, no adverse effects were reported in either group during the intervention period.

## 4. Discussion

The administration of a leucine-enriched essential amino acid supplement for 3 months significantly improved ADL performance and increased muscle mass and strength in older post-stroke patients with sarcopenia. These findings suggest that supplementation, known to stimulate muscle protein synthesis in older adults, effectively mitigated muscle loss in this population.

Resistance training is widely regarded as the standard method for enhancing muscle mass, strength, and physical function. However, this clinical trial focused on isolating the specific effects of targeted nutritional interventions. Existing data on the influence of nutritional supplementation on muscle mass, strength, and physical function in older adults with sarcopenia are limited and contradictory. Bauer et al. [21] reported that nutritional supplements improved muscle mass and chair standability, but did not significantly impact handgrip strength or Short Physical Performance Battery scores. Similarly, Fiataron et al. [22] found that nutritional supplementation alone did not significantly affect muscle mass or physical function. In a recent systematic review and meta-analysis, Yoshimura et al. [5] highlighted the evidence supporting the efficacy of nutritional supplements in improving muscle mass, strength, and physical function; however, the evidence was considered insufficiently robust. Park et al. [23] suggested that a comprehensive rehabilitation intervention combined with BCAA supplementation could be a helpful option during the critical period of post-stroke neurological recovery. The results of this study are consistent with those of previous studies.

This study provides a viable approach to maintain body mass index (BMI) and bone mineral content in stroke patients, addressing a critical issue often overlooked in clinical practice. In particular, preserving bone mineral content during post-stroke rehabilitation may offer potential long-term benefits, such as reducing fracture risk and improving quality of life. Approximately 50% of patients with stroke experience malnutrition [3], a factor that can predict negative post-stroke outcomes (e.g., physical dependency, muscle weakness, extended hospital stays, and decreased quality of life). When combined with malnutrition, sarcopenia may further worsen stroke prognosis [24,25]. Moreover, clinical outcomes in stroke patients may depend significantly on the pre-stroke nutritional status and subsequent nutritional management. Accordingly, further clinical and foundational research is essential to clarify the etiology of stroke-related sarcopenia and to develop effective preventive and therapeutic strategies, including nutritional support.

The quality of protein intake is also a critical determinant of muscle mass and strength [5,26]. Leucine, a vital regulator of postprandial muscle protein synthesis, functions via the mTOR pathway [27]. Signal transduction through mTORC1, which plays a central role in muscle growth by enhancing protein translation, is regulated by intracellular amino acid supply [27]. The consumption of supplements, as used in this trial, enhances muscle protein synthesis and reduces post-exercise muscle soreness [13,14]. Therefore, leucine-enriched essential amino acid supplements may help preserve muscle mass and physical function in older adults with sarcopenia and reduced food intake.

To maximize the effects of nutritional intervention, the patients in this study participated in low-intensity resistance training as part of their rehabilitation. Given the physical limitations common among older adults, the sit-to-stand exercise was selected for its ease of execution and the option for assistance if needed. Low-intensity resistance training with slow movements (e.g., sit-to-stand exercises) has been shown to be safe and effective in increasing muscle mass and strength in older adults with frailty [5,28].

This study has some limitations. First, the relatively small sample size, potential selection bias, and single-center design may have limited the generalizability of the findings. Second, the absence of a control group that did not undergo resistance training means that the study did not distinguish the effects of nutritional support from those of exercise alone. Third, we did not collect data on the duration since stroke onset, stroke subtypes, or the presence of diabetes. Although our study primarily focuses on the effects of nutritional supplementation and rehabilitation on stroke patients undergoing rehabilitation regardless of stroke subtype, we note that future studies with a multicenter approach and larger cohorts should explore stroke subtype-specific responses to intervention.

## 5. Conclusions

This study demonstrated that a 3-month intervention combining a leucine-enriched amino acid supplement taken within 1 h of low-intensity resistance training effectively increased BMI and preserved bone mineral content in post-stroke patients with sarcopenia without negatively affecting renal function. This approach is effective and safe, particularly for elderly patients with physical limitations in standard stroke care. Further research is warranted to investigate the potential of this nutritional supplement alongside other rehabilitation strategies to improve outcomes in stroke patients with sarcopenia.

## Figures and Tables

**Figure 1 nutrients-16-04264-f001:**
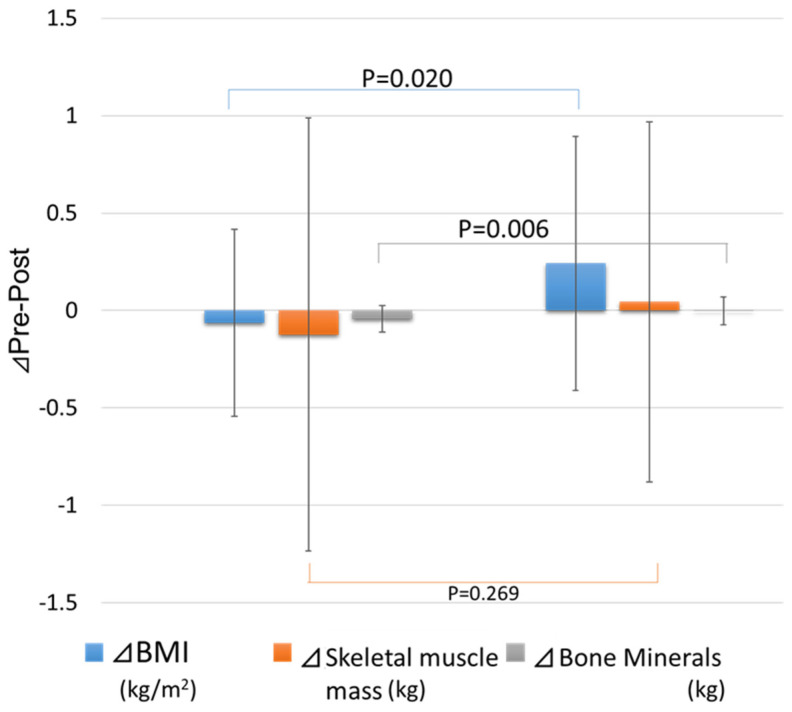
Changes in body composition after the intervention. Mann–Whitney U test.

**Table 1 nutrients-16-04264-t001:** Baseline clinical characteristics of the study participants and between-group differences with and without intervention.

Variables at Baseline	Rehabilitation Alone (n = 31)	Rehabilitation + Nutrition (n = 29)	*p*-Value
Median age, years (IQR)	72 (66–77)	70 (65–78)	0.796
Sex, n (%)			0.320
Male	22 (71.0)	17 (58.6)	
Female	9 (29.0)	12 (41.4)	
Body mass index, kg/m^2^ (SD)	24.3 ± 3.1	23.8 ± 3.4	0.437
Systolic blood pressure (mmHg)	126.6 ± 9.1	125.1 ± 9.1	0.543
Diastolic blood pressure (mmHg)	73.2 ± 7.7	71.6 ± 8.9	0.426
Pulse rate (min)	73.5 ± 10.2	72.5 ± 9.2	0.832
White blood cell count (/µL)	5640.0 ± 1484.3	5872.1 ± 2037.2	0.564
Hemoglobin (g/dL)	14.2 ± 1.9	13.9 ± 1.4	0.153
Platelet count (×10^4^/µL)	23.6 ± 5.5	22.1 ± 6.8	0.283
Serum aspartate transaminase (U/L)	22.5 ± 5.2	22.6 ± 5.4	0.733
Serum alanine transaminase (U/L)	19.4 ± 8.6	20.7 ± 10.5	0.744
Serum γ-glutamyl transpeptidase (U/L)	28.8 ± 11.8	40.3 ± 45.3	0.894
Serum creatinine (mg/dL)	0.8 ± 0.2	0.8 ± 0.2	0.976
eGFR-creatinine (mL/min/1.73 m^2^)	81.9 ± 20.1	84.6 ± 29.3	0.756
Serum cystatin C (mg/L)	1.1 ± 0.1	1.1 ± 0.2	0.882
eGFR-cystatin C (mL/min/1.73 m^2^)	65.9 ± 10.8	67.2 ± 18.4	0.690
Serum uric acid (mg/dL)	5.3 ± 1.4	5.3 ± 1.2	0.745
Serum albumin (g/dL)	4.2 ± 0.2	4.2 ± 0.3	0.846
Serum prealbumin (mg/dL)	24.4 ± 5.9	24.1 ± 4.8	0.982
Serum retinol-binding protein (mg/dL)	2.9 ± 0.9	2.8 ± 0.7	0.651
Geriatric nutritional risk index	103.2 ± 3.7	102.9 ± 4.7	0.931
Serum C-reactive protein (mg/dL)	0.2 ± 0.3	0.1 ± 0.1	0.768
Urine protein creatinine ratio (g/gCr)	0.01 ± 0.04	0.02 ± 0.04	0.292
Serum NT-proBNP (pg/mL)	150.8 ± 435.8	127.9 ± 355.6	0.680
Skeletal muscle mass (kg)	22.9 ± 4.7	21.2 ± 4.0	0.176
Bone minerals (kg)	2.3 ± 0.4	2.2 ± 0.3	0.135
Body fat mass (kg)	20.9 ± 7.0	19.5 ± 7.3	0.520
Basal metabolism rate (kcal)	1292.7 ± 169.2	1228.0 ± 140.0	0.128
Muscle strength			
Hand grip (kgf) Paretic side	13.8 ± 9.8	11.7 ± 10.5	0.342
Hand grip (kgf) Non-paretic side	29.2 ± 9.4	27.3 ± 7.3	0.336
Leg extensor torque (N) Paretic side	175.3 ± 111.8	163.7 ± 91.6	0.673
Leg extensor torque (N) Non-paretic side	235.9 ± 158.4	254.5 ± 102.8	0.222
Motor function			
Confortable 10-m walking speed (m/s)	0.94 ± 0.96	0.86 ± 0.72	0.791
Maximum 10-m walking speed (m/s)	1.22 ± 1.18	1.05 ± 0.92	0.438
2-min walking distance (m)	96.6 ± 51.8	85.4 ± 48.5	0.474
30-s Chair test (times)	12.8 ± 6.7	11.4 ± 4.3	0.795
Functional Independence Measure			
Total	114.6 ± 18.5	114.8 ± 11.1	0.509
Motor	81.9 ± 13.0	81.7 ± 9.2	0.302
Cognitive	32.7 ± 6.0	33.1 ± 3.7	0.362
Sarcopenia	16 (51.6%)	17 (58.6%)	0.974
Severe sarcopenia	5 (16.1%)	6 (20.7%)	0.643

NT-proBNP, N-terminal-pro-brain natriuretic peptide.

**Table 2 nutrients-16-04264-t002:** Changes in the clinical characteristics of the study participants and between-group differences with and without intervention.

Variables at Baseline	Rehabilitation Alone (n = 31)	Rehabilitation + Nutrition (n = 29)	*p*-Value
ΔBody mass index, kg/m^2^ (SD)	−0.06 ± 0.48	0.24 ± 0.65	0.020
ΔSystolic blood pressure (mmHg)	−4.2 ± 13.4	1.6 ± 12.2	0.139
ΔDiastolic blood pressure (mmHg)	−2.3 ± 9.1	−1.7 ± 7.8	0.779
ΔPulse rate (min)	0.6 ± 7.2	−0.3 ± 5.8	0.732
ΔWhite blood cell count (/uL)	144.8 ± 833.9	−185.2 ± 1211.6	0.294
ΔHemoglobin (g/dL)	−0.01 ± 0.53	0.02 ± 0.69	0.959
ΔPlatelet count (×10^4^/uL)	−1.0 ± 2.9	−0.1 ± 2.8	0.141
ΔSerum aspartate transaminase (U/L)	0.35 ± 3.86	0.52 ± 4.14	0.870
ΔSerum alanine transaminase (U/L)	0.97 ± 4.76	−0.03 ± 6.54	0.389
ΔSerum γ-glutamyl transpeptidase (U/L)	1.23 ± 6.62	−0.86 ± 14.18	0.807
ΔSerum creatinine (mg/dL)	−0.01 ± 0.07	−0.02 ± 0.08	0.553
ΔeGFR-creatinine (mL/min/1.73 m^2^)	1.32 ± 6.97	1.44 ± 10.73	0.668
ΔSerum cystatin C (mg/L)	−0.01 ± 0.11	−0.03 ± 0.09	0.446
ΔeGFR-cystatin C (mL/min/1.73 m^2^)	1.69 ± 7.18	2.22 ± 6.52	0.487
ΔSerum uric acid (mg/dL)	−0.04 ± 0.69	−0.29 ± 0.75	0.269
ΔSerum albumin (g/dL)	−0.03 ± 0.20	−0.03 ± 0.25	0.562
ΔSerum prealbumin (mg/dL)	0.22 ± 2.48	−0.03 ± 2.68	0.713
ΔSerum retinol-binding protein (mg/dL)	0.08 ± 0.31	−0.03 ± 2.68	0.088
ΔGeriatric nutritional risk index	−0.03 ± 0.54	0.10 ± 0.29	0.139
ΔSerum C-reactive protein (mg/dL)	0.02 ± 0.27	0.01 ± 0.11	0.743
ΔUrine protein creatinine ratio (g/gCr)	0.01 ± 0.10	0.02 ± 0.13	0.146
ΔSerum NT-proBNP (pg/mL)	−22.4 ± 95.7	7.4 ± 99.7	0.383
ΔSkeletal muscle mass (kg)	−0.06 ± 0.70	−0.01 ± 0.54	0.269
ΔBone minerals (kg)	−0.04 ± 0.07	−0.01 ± 0.07	0.006
ΔBody fat mass (kg)	−0.12 ± 1.38	0.95 ± 2.42	0.076
ΔBasal metabolism rate (kcal)	−6.6 ± 19.8	1.5 ± 20.5	0.092
ΔMuscle strength			
ΔHand grip (kgf) Paretic side	0.94 ± 3.70	2.60 ± 8.12	0.834
ΔHand grip (kgf) Non-paretic side	−3.15 ± 11.68	−1.28 ± 8.78	0.749
ΔLeg extensor torque (N) Paretic side	28.02 ± 76.61	27.94 ± 58.74	0.587
ΔLeg extensor torque (N) Non-paretic side	14.96 ± 79.20	34.76 ± 59.04	0.201
ΔMotor function			
ΔConfortable 10-m walking speed (m/s)	0.02 ± 0.12	0.01 ± 0.15	0.403
ΔMaximum 10-m walking speed (m/s)	0.02 ± 0.15	0.01 ± 0.13	0.857
Δ2-min walking distance (m)	3.04 ± 11.30	3.38 ± 15.89	0.931
Δ30-s Chair test (times)	0.40 ± 2.08	1.26 ± 2.78	0.603
ΔFunctional Independence Measure			
ΔTotal	−0.1 ± 1.6	0.1 ± 1.1	0.584
ΔMotor	0.2 ± 0.6	0.1 ± 1.1	0.184
ΔCognitive	0.0 ± 0.0	0.0 ± 0.0	1.000

NT-proBNP, N-terminal-pro-brain natriuretic peptide.

## Data Availability

Original data are available with the corresponding author but not archived in the database elsewhere.

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
