# Peer review of "Effects of Nutritional Support with a Leucine-Enriched Essential Amino Acid Supplement on Body Composition, Muscle Strength, and Physical Function in Stroke Patients Undergoing Rehabilitation"

_nutrients, 2024, doi:10.3390/nu16244264_

Round 1

Reviewer 1 Report

Comments and Suggestions for Authors

The article is valuable, innovative and well written. Most of my suggestions are minor changes. The methods section and the description of the patient group need major changes.

Abstract: 

1. In response to the question whether the stroke group was homogeneous or mixed (i.e. included ischemic, hemorrhagic or lacunar strokes),

DOI: 10.1038/s41598-020-69831-0

2. The methodology used to quantify significant differences between groups in variables such as BMI and mineral loss should be clearly stated.

Introduction:

3. The discussion on sarcopenia should be supplemented with an analysis of the incidence rate of albuminuria and general malnutrition. 

DOI: 10.1186/s12877-022-02916-9 

DOI: 10.3389/fneur.2022.744945

Material and methods: 

4. Again, it would be beneficial to determine what type of stroke it is. 

5. Please provide a detailed description of the inclusion and exclusion criteria. 

6. What methods were used to diagnose stroke? 

7. How many patients had diabetes? 

8. It would be beneficial to include indicators of the nutritional status of the patients studied. 

DOI: 10.1016/j.transproceed.2024.03.012 

Conclusions :

9. Please provide the time of intervention and the leucine content of the supplement.

Reviewer 2 Report

Comments and Suggestions for Authors

The manuscript "Effects of nutritional support with a leucine-enriched essential amino acid supplement on body composition, muscle strength, and physical function in stroke patients undergoing rehabilitation" presents an insightful investigation into the impact of a leucine-enriched essential amino acid supplementation combined with resistance training on post-stroke patients with sarcopenia. The research design, involving a randomized controlled trial with well-defined intervention and outcome measures, offers valuable evidence on the potential benefits of combining nutritional support with physical rehabilitation. While the study is commendable for its clinical relevance and contribution to stroke care, several areas require enhancement to improve the clarity, depth, and interpretability:

1. (Lines 11–31): Simplify the abstract by reducing repetitive statements and emphasizing key results. For example, the distinction between groups in changes in BMI and bone mineral content can be highlighted briefly.

2. (Lines 63–69): Clarify the hypothesis and objectives. Instead of "evaluate the efficacy and safety," specify what constitutes "efficacy" and "safety" in measurable terms (e.g., changes in specific functional markers).

3. (Lines 35–69): Strengthen the introduction by better linking malnutrition and sarcopenia to stroke outcomes. Cite newer references to highlight recent advances in understanding post-stroke sarcopenia.

4. (Lines 58–67): Expand on why leucine-enriched supplementation is superior to other interventions. Discuss mechanisms in greater detail, such as their role in mTOR signaling.

5. (Lines 79–91): Provide more data on participants, such as duration since stroke onset, stroke subtypes, and baseline nutritional status, to clarify the population studied.

6. (Lines 99–123): Include a detailed justification for the chosen supplement composition (Reha-Time Jelly®). Explain why this specific formulation was used over others.

7. (Lines 105–114): Specify the intensity and duration of each exercise within the rehabilitation program. Including guidelines or references for the exercise protocol would improve reproducibility.

8. (Lines 135–154): Add justification for using specific measures like NT-proBNP, which may not directly relate to sarcopenia or functional recovery.

9. (Lines 165–220): Present results more systematically by separating primary (e.g., muscle strength, BMI) and secondary (e.g., eGFR, NT-proBNP) outcomes. Use subheadings for clarity.

10. (Figure 1 and Table 2, Lines 187–199): Ensure that figures and tables are self-explanatory. For Figure 1, explain the clinical significance of changes in BMI and bone mineral content.

11. (Lines 201–244): Expand on the implications of preserving bone mineral content in post-stroke rehabilitation. Discuss potential long-term benefits of the intervention on fracture risk or quality of life.

12. (Lines 241–245): Extend the limitations to include potential biases due to the small sample size and single-center design. Suggest future studies with a multicenter approach and larger cohorts.

13. (Lines 246–252): Strengthen the conclusion by directly tying findings to clinical practice. Highlight the potential integration of nutritional supplementation in standard stroke care.

Round 2

Reviewer 1 Report

Comments and Suggestions for Authors

The authors answered the questions asked and supplemented the content of the article. when correcting articles next time it is worth marking the changes in color, which makes it easier for the reviewer to check.

I wish you much success

Reviewer 2 Report

Comments and Suggestions for Authors

The authors have made revisions based on my previous comments and provided point-by-point responses that satisfy all my concerns. In such circumstances, I recommend acceptance of publication.